# PenDA, a rank-based method for personalized differential analysis: Application to lung cancer

**Magali Richard**[1]*, **Clémentine Decamps**[1], **Florent Chuffart**[2], **Elisabeth Brambilla**[3], **Sophie Rousseaux**[2], **Saadi Khochbin**[2], **Daniel Jost**[1,4]*

**1** Univ Grenoble Alpes, CNRS, Grenoble INP, TIMC-IMAG, Grenoble, France, **2** CNRS UMR 5309, Inserm U1209, Univ Grenoble Alpes, Institute for Advanced Biosciences, Grenoble, France, **3** CHUGA, Inserm U1209, Univ Grenoble Alpes, Institute for Advanced Biosciences, Grenoble, France, **4** University of Lyon, ENS de Lyon, Univ Claude Bernard, CNRS, Laboratory of Biology and Modelling of the Cell, Lyon, France

\* magali.richard@univ-grenoble-alpes.fr (MR); daniel.jost@ens-lyon.fr (DJ)

**Data Availability Statement:** LUAD and LUSC expression data were downloaded from The Cancer Genome Atlas program (https://portal.gdc.cancer.gov/). Grenoble hospital expression data are accessible in the study GSE30219. Rmarkdown

## Abstract

The hopes of precision medicine rely on our capacity to measure various high-throughput genomic information of a patient and to integrate them for personalized diagnosis and adapted treatment. Reaching these ambitious objectives will require the development of efficient tools for the detection of molecular defects at the individual level. Here, we propose a novel method, PenDA, to perform Personalized Differential Analysis at the scale of a single sample. PenDA is based on the local ordering of gene expressions within individual cases and infers the deregulation status of genes in a sample of interest compared to a reference dataset. Based on realistic simulations of RNA-seq data of tumors, we showed that PenDA outcompetes existing approaches with very high specificity and sensitivity and is robust to normalization effects. Applying the method to lung cancer cohorts, we observed that deregulated genes in tumors exhibit a cancer-type-specific commitment towards up- or down-regulation. Based on the individual information of deregulation given by PenDA, we were able to define two new molecular histologies for lung adenocarcinoma cancers strongly correlated to survival. In particular, we identified 37 biomarkers whose up-regulation lead to bad prognosis and that we validated on two independent cohorts. PenDA provides a robust, generic tool to extract personalized deregulation patterns that can then be used for the discovery of therapeutic targets and for personalized diagnosis. An open-access, user-friendly R package is available at https://github.com/bcm-uga/penda.

## Author summary

The hopes of precision medicine rely on our capacity to measure individual molecular information for personalized diagnosis and treatment. These challenging perspectives will be only possible with the development of efficient methodological tools to identify patient-specific molecular defects from the many precise molecular information that one can access at the single-individual, single tissue or even single-cell levels. Such methods

vignettes (S1 & S2 Texts) for reproducing all figures and tables in this paper can be found in a R package named penda (https://github.com/bcm-uga/penda). Preprocessed data used to generate the figures can be found at http://membres-timc.imag.fr/Magali.Richard/publication.html: tcga_data_ctrl.rds, tcga_data_case.rds, tcga_exp_grp.rds, grenoble_data_ctrl.rds, grenoble_data_case.rds, grenoble_exp_grp.rds.

**Funding:** The research leading to these results was supported by ITMO Cancer (Plan Cancer 2014-2019, Biologie des Systèmes n°BIO2015-08) [EB, SK, DJ] and University Grenoble-Alpes via the Grenoble Alpes Data Institute [MR] and the SYMER program [SK, DJ] (which are funded by the French National Research Agency under the "Investissements d'Avenir" program ANR-15-IDEX-02). SK acknowledges additional funding from Plan Cancer ASC16079CSA, Pitcher, LIFE program of University Grenoble Alpes (ANR-15-IDEX-02), Fondation ARC "Canc'air" (RAC16042CLA) and project PGA1 RF20190208471. The funders had no role in study design, data collection and analysis, decision to publish, or preparation of the manuscript.

**Competing interests:** The authors have declared that no competing interests exist.

will provide a better understanding of disease-specific biological mechanisms and will promote the development of personalized therapeutic strategies. Here we describe a novel method, named PenDA, to perform differential analysis of gene expression at the individual level. Based on a realistic benchmark of simulated tumors, we demonstrated that PenDA reaches very high efficiency in detecting sample-specific deregulated genes. We then applied the method to two large cohorts associated with lung cancer. A detailed statistical analysis of the results allowed to isolate genes with specific deregulation patterns, like genes that are up-regulated in all tumors or genes that are expressed but never deregulated in any tumors. Given their specificities, these genes are likely to be of interest in therapeutic research. In particular, we were able to identified 37 new biomarkers associated to bad prognosis that we validated on two independent cohorts.

## Introduction

General medicine still largely relies on detecting diseases after the apparition of symptoms and on curing them with generic treatments. However, many studies have highlighted how the natural genetic or genomic diversities observed in a population, as well as patient history, or environment exposure, may strongly affect diseases risks, prognoses and responses to treatments [1,2]. This is particularly critical for cancer, where each individual tumor may be viewed as an independent disease, with specific and variable responses to generic therapeutic treatments [3]. Recently, thanks to the development of cheap and robust next-generation sequencing techniques, getting better insights into inter-individual heterogeneities was made possible by the analyses of large cohorts of patients. This led to the identification of individual molecular signatures or biomarkers associated with better prognosis, or better response to targeted treatment [4–6]. This new knowledge paves the way to precision and personalized medicine where the genetic, genomic, and molecular information of each patient will be integrated to develop personalized diagnosis and treatment [2,3]. However, such challenging perspectives will be only possible with the concomitant development of efficient and robust methodological tools that allow the identifications of molecular defects or deregulation patterns at the individual level.

Many statistical or bioinformatic methods do already exist to identify deregulated genes at the population level. For example, in the context of gene expression, standard methods like DESeq2 [7], edgeR [8] or limma [9] are designed and routinely used to identify genes that are differentially expressed *in average* between two groups of patients [10]. These methods are usually based on modelling of the data distribution and statistical testing for differential expression (fold change analysis). While valuable to detect consistent *typical* deregulation patterns, such analyses do not provide precise information at the individual level. In addition, these global methods are usually very sensitive to batch effects that, without corrections, may lead to false discoveries or to confound important subpopulation effects [11]. Prior application of normalization routines to the investigated samples are used to mitigate such technical biases, but improper normalization may still perturb the biological signal [12,13].

Novel methods, robust to technical interference, are therefore needed to capture specific, individual data. Few promising techniques already allow to extract interpretable information from personalized omics data (see [14] for a review). Rankcomp [15,16] uses pairs of genes with a stable, relative order in a reference dataset to infer deregulated genes in individual samples [17–19]. This method, based on ranking, avoids the problem of normalization between samples, but results in very high false discovery rates (above 20%, see Methods). Alternative

methods, like DEGseq [20], NOISeq [21] or Gfold [22], exploit paired samples from the same patient (one control versus one malignant) to perform differential analysis. However, such matched samples are usually rare (for example, in the case of cancer, a single sample from the tumorous biopsy is usually available for one patient). Above all, it is not clear if the variabilities observed between paired samples are due to actual deregulation, to intrinsic inter-sample heterogeneities, or to technical biases. For example, in lung cancer, correlations between paired tumorous and normal samples are similar than between tumors of two different patients, and are only slightly higher than between a tumorous sample and an unmatched normal tissue (Fig 1A).

To overcome all these limitations, we developed PenDA, for Personalized Differential Analysis, a rank-based method, robust to batch or normalization effects, that uses information extracted from a reference dataset to infer the deregulation status of genes in individual samples of interest.

For illustrating the power of the method, we focused on lung cancer, which is the first cause of cancer-related death world-wide [23] and represents a major public health issue. In particular, we studied two datasets provided by The Cancer Genome Atlas (TCGA) for two of the most common histologies of non-small-cell lung cancers (NSCLCs): adenocarcinoma (ADC [24]) and squamous cell carcinoma (SQCC [25]). Clinical implications, gene expression patterns and DNA mutation landscapes are largely distinct between both histologies even if some pathways are similarly altered [26]. Their mutation rates are unusually high compared to other lung cancers and molecular heterogeneity is important [24,25,27]. This molecular heterogeneity translates into a complex landscape of deregulation of gene expression [28,29]. Previous analyses of molecular abnormalities occurring in a large proportion of patients have already led to the development of biomarkers for target therapy [30] and for prognostic signatures [31] but it still remains an important biomedical priority [32]. More generally, observations of morphological, histological or molecular defects led to the classifications of ADC and SQC into various subtypes [24,25,28,33–36]. For example, ADC is generally classified into three subtypes according to transcriptional and histopathological data: terminal respiratory unit (TRU or bronchoid), proximal inflammatory (PI or squamoid) and proximal proliferative (PP or magnoid). These subtypes differ by gene expression but also by clinical behaviors like the stage-specific survival [24,36,37]. Recently, Chen et al [27] combined various molecular information (DNA methylation, copy number alteration, mRNA, miRNA and protein expression) to define 6 molecular subtypes of ADC that partially overlap with the standard classification and that show correlation with survival rate, immune profiles or cigarette exposure. By using our method PenDA on the TCGA datasets for ADC and SQC, we illustrated how personalized differential analysis can bring additional information compared to previous studies about inter-individual heterogeneity, can help to find gene classifiers for molecular subtypes and may be used to infer biomarkers related to prognosis.

## Results

### A robust algorithm to infer if genes are differentially regulated in individual samples

**Description of the method.** PenDA is a rank-based method that allows to infer if the expression of any gene in a given sample of interest is deregulated compared to a set of reference samples (see Methods for details). The fundamental assumption behind the algorithm is that a gene is seen as deregulated in an individual sample if its local ordering compared to other genes with similar expressions is perturbed, as similarly stated by the RankComp method [15]. Briefly, PenDA starts by inferring a reference of relative ordering in control samples: for

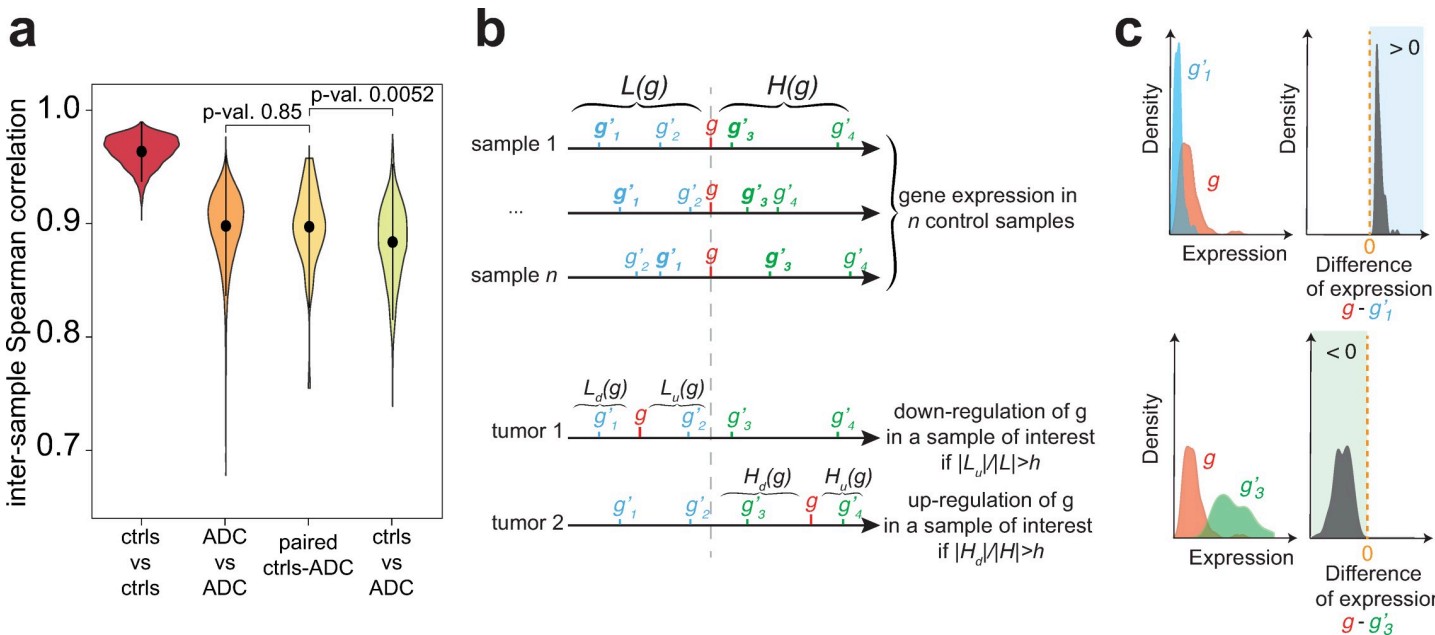

**Fig 1. The PenDA method.** (a) Violin-plots for the distributions of Spearman correlation between two samples taken from the TCGA database on lung adenocarcinoma: between two non-tumorous samples (ctrls vs ctrls, n = 4,656 pairs), between two tumorous samples (ADC vs ADC, n = 103,285), between paired normal and tumorous samples (paired ctrls-ADC, n = 48), and between unpaired controls and tumors (ctrls vs ADC, n = 44,135). Shown p-values correspond to Wilcoxon tests. (b) Basic scheme depicting the PenDA method. (Top) For each gene $g$, the algorithm infers sets of genes whose expressions are always lower ($L(g)$) or higher ($H(g)$) than that of $g$ in a pool of control, reference samples. (Bottom) In a given individual (tumor) sample, $g$ is viewed as deregulated if its relative ordering with genes in the $L(g)$ and $H(g)$ lists is modified. (c) Examples of genes in the $L$ ($g'_1$, top) or $H$ ($g'_3$, bottom) lists of a gene $g$. While the individual distributions of gene expression in the control samples may overlap (left), the distribution of the difference in gene expression in controls (right) is always positive or negative for genes in $L$ and $H$ lists respectively.

every gene $g$, it constructs two lists $L(g)$ and $H(g)$ of genes whose expression is lower and higher respectively than that of $g$ in almost all the samples of a given reference dataset (Fig 1B top and 1C). To avoid comparison with genes having very different expression levels and to increase sensitivity of the method, lists $L(g)$ and $H(g)$ are then limited to the subset of $l$ genes whose expression in control samples are closest to $g$. Finally, for a given sample of interest, PenDA scans every gene $g$ to determine if it might be up- or down-regulated in that sample. This step is performed by considering the number of genes $L_u(g)$ (respectively $H_d(g)$) in $L(g)$ (resp. $H(g)$) in the studied case whose relative ordering to $g$ has changed compared to controls (Fig 1B bottom). If the proportion of such genes with a modified order ($|L_u(g)|/|L(g)|$ or $|H_d(g)|/|H(g)|$) exceeds a given threshold $h$, the gene $g$ is detected as deregulated. It has to be noted that a change of ordering between $g$ and a gene $g'$ of $L(g)$ and $H(g)$ might be caused by the deregulation of $g'$ and not necessary by that of $g$. To limit the consequences of this effect on the detection of deregulation, PenDA iteratively applies the previous scheme until convergence by excluding at each iteration the current set of deregulated genes from every $L$ and $H$ lists (S1 Fig). In the cases where the $L(g)$ or $H(g)$ lists are empty, we used the percentile method (see Methods for details) to evaluate the deregulation of $g$ (S2 Fig).

**Impact of method parameters and of the dataset properties on performance.** To test and validate our method, we generated a realistic simulated dataset where we controlled the identity of deregulated genes and the direction (up or down) of deregulation. Based on the RNA-seq profiles of 18,000 genes in normal and tumorous samples of two lung cancer cohorts (adenocarcinoma: ADC, squamous cells: SQCC) of the TCGA database [24,25], we simulated 10 tumorous samples each having on average 30% of deregulated genes (see Methods for details). Note that to avoid any bias in the analysis, simulations were not based on the same

principle that governed the PenDA method, ie, the relative order of gene expressions. Rather, each *in silico* tumor was generated by randomly choosing a normal sample and a list of deregulated genes was randomly assigned. Then, the perturbed gene expressions of these genes were obtained by adding to the normal levels random values typical of the differences in gene expression between tumorous and normal samples as observed in the actual dataset.

We first aimed at testing the method on this dataset by varying the two parameters of the algorithm: $l$ the restricted size of the $L(g)$ and $H(g)$ lists, and $h$ the detection threshold based on the $|L_u(g)|/|L(g)|$ and $|H_d(g)|/|H(g)|$ ratios (see above). We used the 97 non-tumorous lung samples of the TCGA dataset to determine $L(g)$ and $H(g)$ and then, apply the PenDA method to the 10 simulations. By varying $h$ from 0 to 1, we built a ROC curve (true positive rate TPR vs false positive rate FPR) for different $l$ values (Fig 2A). We observed that all the curves are well above the line of no-discrimination (dashed grey line), reaching simultaneously high sensitivity and high specificity. Using the maximal value of informedness (TPR-FPR) as a summary statistic of the ROC curve, we observed that the method reached an optimal prediction efficiency for $l \sim 10$–$100$. For too short lists, finite size effects dominate and decrease the predictive power. For very large lists, $L(g)$ and $H(g)$ contain many genes whose expressions are very far

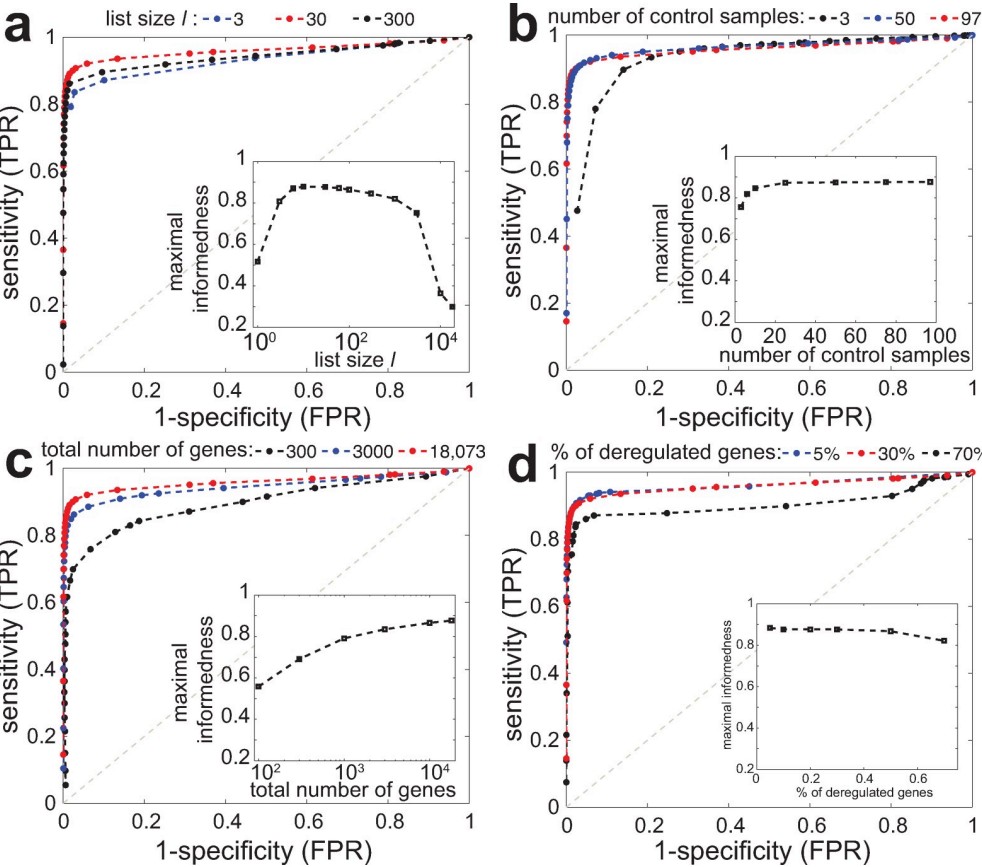

**Fig 2. Parameter analysis and predictive power.** ROC curves (true positive rate TPR vs false positive rate FPR) of the PenDA method on simulated datasets. The curves were obtained by varying the proportion threshold $h$ for various values of other method parameters or of properties of the investigated dataset. Insets show the maximal informedness that represent the maximal value of the difference TPR-FPR computed for each ROC curve. (a) Effect of the maximal size $l$ of $L$ and $H$ lists. (b) Impact of the number of control samples used to infer the $L$ and $H$ lists. (c) Effect of the total number of genes in the dataset. (d) Impact of the proportion of deregulated genes in the tumorous samples.

from *g*. Thus, if *g* is weakly or mildly deregulated, these genes will keep their relative position compared to *g*, leading to a loss in sensitivity. In the next, we imposed *l = 30*.

We then evaluated how PenDA performance depends on the intrinsic properties of the investigated datasets. We determined *L(g)* and *H(g)* using different numbers of non-tumorous samples and run PenDA on the same set of 10 simulations. We observed that the method is very robust regarding the size of the reference datasets, achieving very high efficiency even for a limited number of control samples (Fig 2B). Next, we kept the reference pool fixed but varied the number of investigated genes from 100 to 18,000 and applied PenDA to the simulated dataset restricted to the corresponding limited set of genes (Fig 2C). We remarked that the reliability of the method is an increasing function of the number of genes, achieving very good performance for numbers higher than ~3,000. Indeed, a large number of genes augments the capacity of *L(g)* and *H(g)* lists to integrate genes that may be sensitive to changes in relative ordering. Finally, we tested the effect of the percentage of deregulated genes in the simulated datasets that may affect the current sizes of *L* and *H* lists during the iterations of the method. Fig 2D showed that the predictive power of PenDA is relatively insensitive to this quantity, performance slightly declining for very high percentage.

All these quantitative analyses illustrate that the method is very robust regarding parameters and dataset properties fine-tuning. In particular, PenDA remains performant even for a small number of reference datasets.

**Comparison with other individual-based methods.** We next sought to compare PenDA with other existing methods that also allow personalized diagnosis of gene deregulation. Using the same set of 10 simulations introduced before, we generated ROC curves (see Methods) for 4 alternative methods (Fig 3): 2 versions of the rank-based method RankComp [15,16], a simple percentile method based on outlier detection and DESeq2 [38], the popular algorithm for detecting differential expression at the population level but used here on an individual basis. We observed that PenDA outperforms these methods, in particular in the limit of high specificity (FPR< = 5%) where PenDA could reach very high sensitivity (TPR> = 90%) even for a limited number of control samples (Fig 3B). Surprisingly, outcomes of the RankComp methods were very dependent to the number of control samples and even lead to better results for smaller control datasets. Note that basing our definition of deregulation on relative rankings

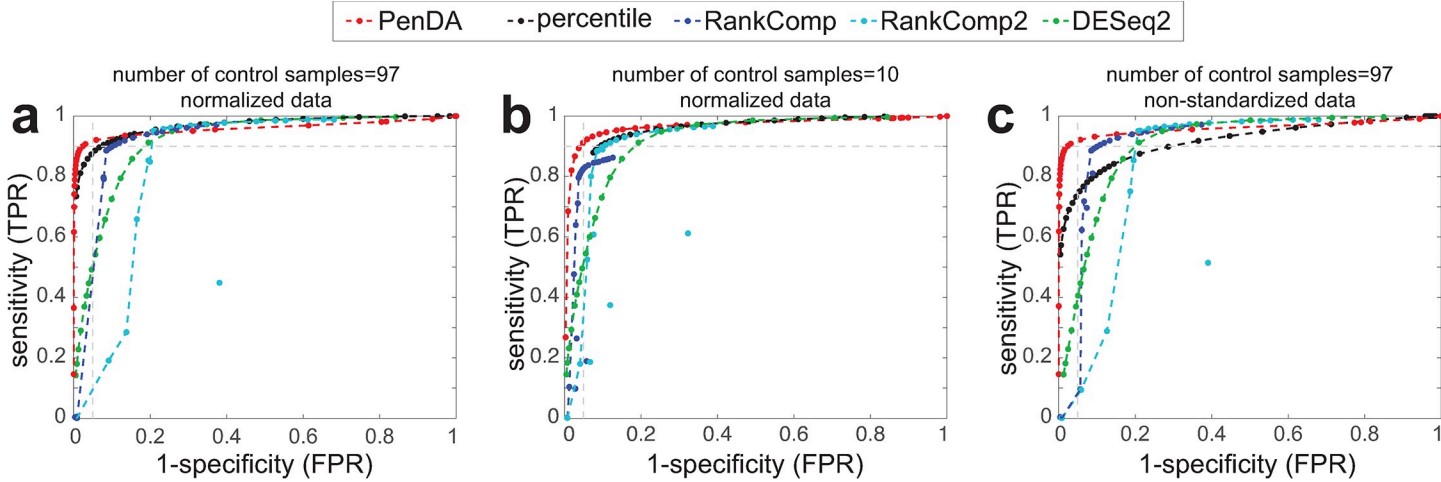

**Fig 3. Comparison with other methods.** (a) ROC curves on the same simulated dataset (normalized data, 97 control samples) as used in Fig 2 for PenDA, a simple percentile-based method, 2 versions of RankComp and DESeq2. (b) As in (a) but reference pool was composed by only 10 control samples. (c) As in (a) but data were not normalized.

limits the sensitivity of PenDA (and RankComp) to batch or normalization effects compared to the percentile method (Fig 3C), DESeq2, thanks to its internal normalization routine, being also robust (S11C Fig).

**The PenDA package.**   The PenDA method is available as a R package at https://github.com/bcm-uga/penda. The *penda* vignette (vignette_penda, S1 Text) runs the PenDA pipeline (S3 Fig) on the samples of interest. It takes as an input two dataframes corresponding to the reference dataset of control samples and the dataset to investigate. It first filters for genes whose expressions are very low in every samples. Then, it computes the *L* and *H* lists from control samples for a given list size *l*. Finally, in every sample, it run the iterative process to infer gene deregulation based on a user-defined threshold *h*. Optionally, the package offers the possibility to find the optimal set of parameters (in particular *h*) best adapted to: (i) the input data and (ii) a user-defined specific maximal false-discovery rate (vignette_simulation, S2 Text). It is based on realistic simulations built on the input dataframes and a ROC analysis, as described in the previous sections. Typically, on a standard personal computer (1 core of 3.6 GHz CPU), construction of *L* and *H* lists takes ~10 sec CPU time for 18,000 genes and 98 controls. Downstream analysis of gene deregulation is slower and requires ~2 min CPU time per analyzed sample.

## Application of the PenDA method to personalized analysis of genetic deregulation in lung cancer

**Overview of gene deregulation in adenocarcinoma and squamous cell carcinoma.**   We evaluated the performances of PenDA on two large cohorts of patients from The Cancer Genome Atlas (TCGA) project representing two of the most common types of non-small-cell lung cancers: lung adenocarcinoma (ADC, ~50%) and lung squamous cell carcinoma (SQCC, ~40%) [39]. Personalized differential analysis was performed on the normalized gene expression data (RNA-seq) of 455 ADC cases and 473 SQCC cases (S1 Table).

We observed that the proportion of deregulated genes per tumor is very variable (Fig 4A), ranging from 3% to 61% of deregulated genes in ADCs (with a mean of 33%, corresponding to 5960 genes) and from 0.4% to 55% of genes deregulated in SQCCs (with a mean of 42%, corresponding to 7659 genes). Analysis of variance revealed a slight effect of tumor stages on the total number of gene deregulations in both ADC and SQCC patients (S4A and S4B Fig). Multiple-comparisons with the Tukey method indicated a significant increase in the number of deregulated genes between an early stage of cancer (stage Ia) and the later stages (stage Ib to stage IV). We consistently observed a higher number of gene down-regulations compared to gene up-regulations in each patient (median ratio down/up of 1.25 in ADCs and of 1.31 in SQCCs). These ratios were invariant across tumor stages (one-way ANOVA non-significant, S4C and S4D Fig).

To test the accuracy of our method, we compared the gene deregulation behavior between ADC and SQCC disease groups. We examined, for each gene, the proportion of tumors where the gene was detected as deregulated within each cohort (Fig 4B and 4C). A two-proportion Z-test was used to compare, for each gene, the observed proportion of deregulation (S2 Table). We identified 5346 genes with a significant variation in down-regulation proportion between ADCs and SQCCs (Fig 4B) and 5616 genes with a significant variation in up-regulation proportion between ADCs and SQCCs (Fig 4C). Gene functional annotation indicated an enrichment in cell division, epidermis development and keratinocyte differentiation in genes specifically up-regulated in SQCCs (S5D Fig). In contrast, genes specifically up-regulated in ADCs display a significant enrichment in glycan processing (S5B Fig). Genes specifically down-regulated in either SQCC or ADC do not display significant enrichment (GO term

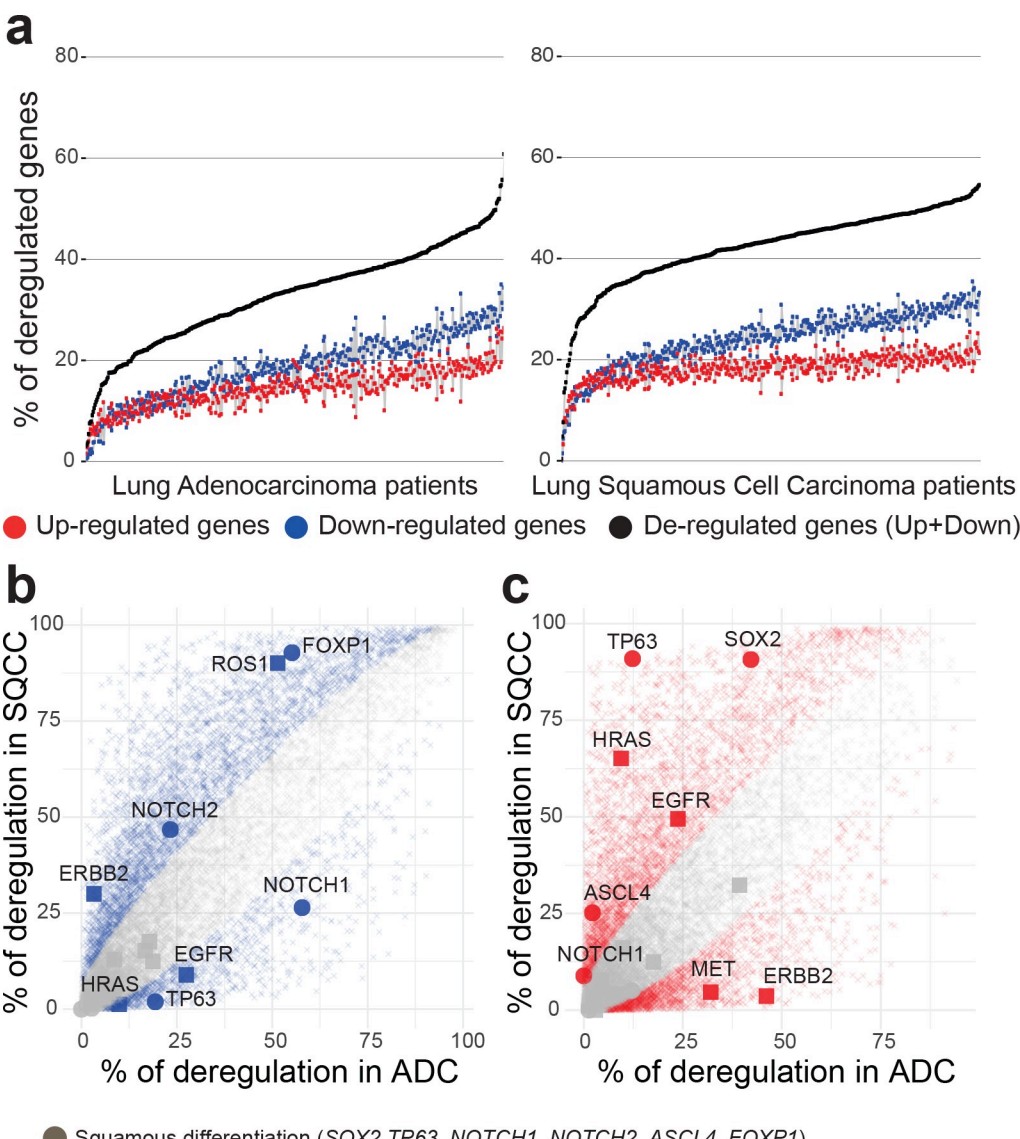

**Fig 4. Overview of genetic deregulation in adenocarcinoma and squamous cell carcinoma.** (a) The percentage of deregulated genes in ADC (left panel) and SQCC (right panel) patients. % of up-regulated genes is indicated in red, % of down-regulated genes is indicated in blue, total % of deregulated genes (up + down) is indicated in black. Patients are ordered by increasing total number of deregulated genes. (b,c) Scatterplot of the percentage of deregulated patients for each gene in the ADC cohort (x-axis) versus deregulated patients percentage in the SQCC cohort (y-axis). Left panel (b) represents downregulation events and right panel (c) represents upregulation events. Colored points represent significant differences between ADC and SQCC cohorts (two-sided two-proportion z-test, p-value < 0.05 after Bonferroni correction for 18143 multiple testing).

significance score < 2). Thus, our method successfully managed to identify biological pathways differentially activated between ADCs and SQCCs.

To illustrate such differential behaviors, we specifically depicted genes belonging to two known pathways involved in cancer progression: the squamous differentiation, that often display somatic alterations in SQCC cancers [25], and the receptor tyrosine kinase (RTK)/RAS/RAF pathway, frequently mutated in ADC cancers [24] (Fig 4B and 4C). In agreement with

previous studies based on population level analysis [40,41], we observed a specific high proportion of up-regulation of SOX 2 and TP63 in SQCCs and of ERBB2 in ADCs. SOX2 is a transcription factor involved in normal squamous cell differentiation, which is frequently amplified in SQCCs [42]. TP63 belongs to the p53 tumor suppressor family, an overexpression of an altered TP63 isoform has been frequently associated with cancer squamous histology [43]. ERRB2 is a member of the epidermal growth factor (EGF) receptor family and is often overexpressed or mutated in ADC [44]. Interestingly, many genes frequently affected by somatic alterations, such as KRAS and EGFR in ADCs[45], exhibit a weaker gene deregulation. In contrast, some genes with a low occurrence of somatic alterations present a strong deregulation frequency in SQCCs, such as FOXP1 or NOTCH1 [25].

Taken together, these results suggest that personalized analysis of both genetic mutations and gene expression variations are required for a full understanding of regulation pathways involved in tumorigenesis.

**Most deregulated genes are committed to specific deregulation patterns.** Recurrent gene deregulations are considered as characteristic features of cancer initiation and progression. To explore the deregulation pattern of each gene, we analyzed their proportion of down-regulation and up-regulation in each cohort (Fig 5A and 5B). Most of the genes that are deregulated in more than ~30% of the patients exhibited a commitment toward up-regulation or down-regulation. For genes deregulated in less that ~30% of the patients, up-regulation and down-regulation are less constrained. Interestingly, ~ 5% of the genes that are either down or up-regulated in more than 30% of both SQCCs and ADCs display antagonistic commitment (S6 Fig). Thus, while the orientation of the deregulation commitment (towards up or down regulation) is generally conserved between ADC and SQCC, in some cases, it may be inverted.

We then decided to quantify extreme single gene deregulation frequencies using a one sample t-test in which we compared the mean deregulation of each gene to the mean deregulation of all genes. Using this approach, we were able to identify genes with specific deregulation patterns, that we defined as super-conserved (SC, genes almost never deregulated), super-up-regulated (SU, genes almost systematically up-regulated) and super-down-regulated (SD, genes almost systematically down-regulated) (S3 Table). While some of the genes with a 'super' regulation pattern are common to ADCs and SQCCs cancers, we observed that a significant proportion of them are specific to a given histology (Fig 5C). Functional profiling indicated that SQCCs SU genes are enriched in cell cycle processes, DNA replication and keratinocyte differentiation. Interestingly, a significant proportion of SQCCs and ADCs SD genes are related to angiogenesis and signal transduction processes (S7 Fig).

As an illustration of the 'super' regulation patterns, we examined more closely three characteristic genes: the SC gene *CAPS*, the SU gene *ESRP1* and the SD gene *RILPL2*. *CAPS* encodes for a calcium binding protein, ESPR1 is an epithelial cell-type-specific splicing regulator and RILPL2 is a rab-interacting lysosomal protein. In Fig 5D–5F, we plotted for these three genes the distribution of gene expression (normalized RNA-seq counts) within the control dataset, the ADC and the SQCC cohorts. Interestingly, for the *CAPS* gene, we do observe a difference in mean expression of the gene between cancer tissues and control whereas no differential expression was detected at individual level. Similarly, expression distributions of *ESRP1* and *RILPL2* genes in ADC and SQCC cohorts partially overlap with their respective distributions in control samples. However, our method identified deregulation in almost all patients of both ADC and SQCC cohorts, indicating that these two genes are committed to specific deregulation pattern during tumorigenesis.

These examples illustrate the power of individual-based approaches compared to population based-approaches. Indeed, extreme single gene deregulation frequencies detection is only possible when individual variations are considered. Those results clearly indicate that a small

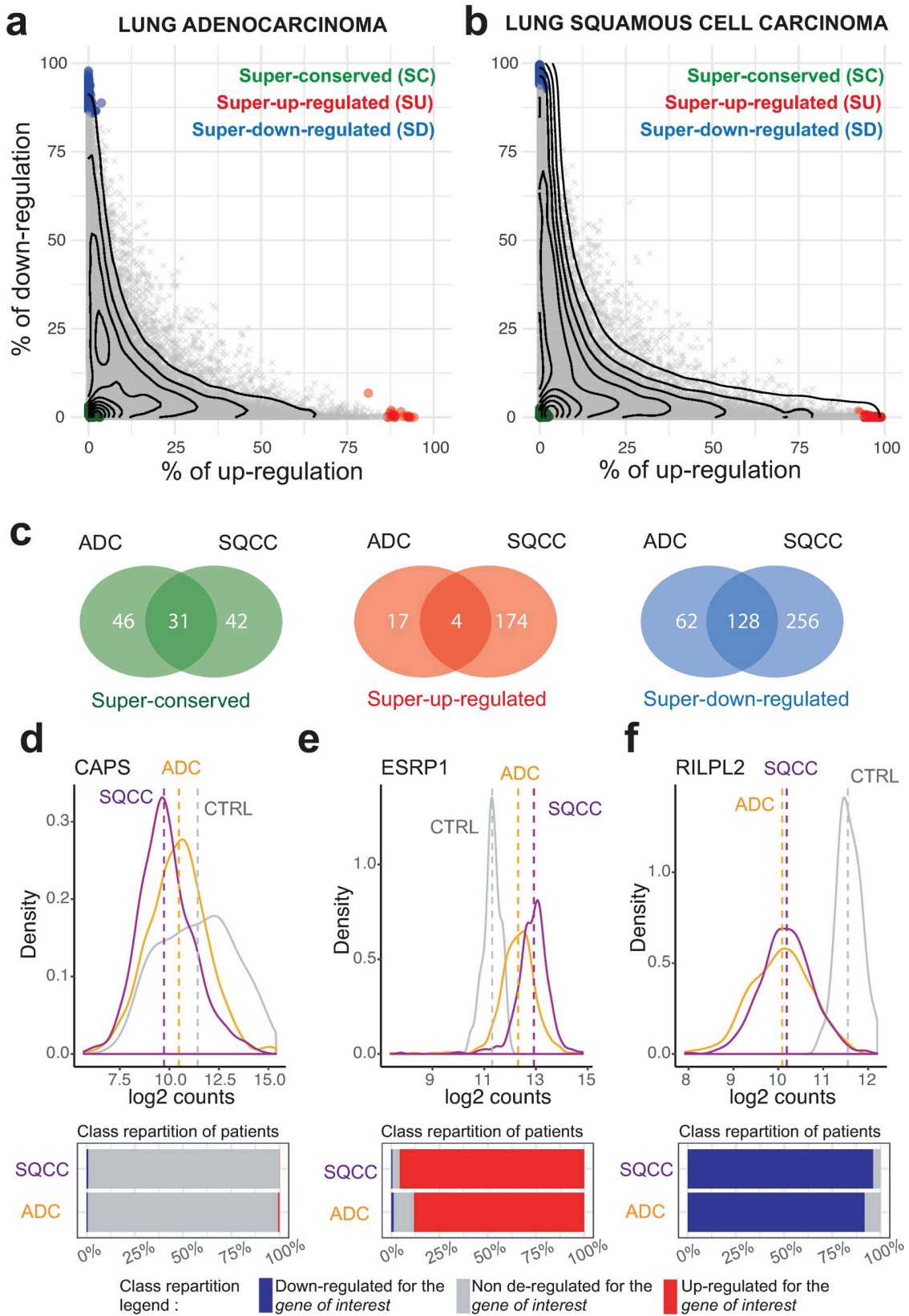

**Fig 5. The gene deregulation pattern.** (a-b) Scatterplots of the percentage of up-regulated versus down-regulated patients in the ADC (left panel) and SQCC (right panel) cohorts. Each dot corresponds to one gene. The x-axis indicates the percentage of up-regulation within the cohort, the y-axis indicates the percentage of down-regulation within the cohort. The contour lines correspond to the density of genes. Genes that are significantly differentially expressed at the individual level (t-statistic, q-value < 0.05) are represented using the following color code: green genes are super-conserved (SC), blue genes are super-down-regulated (SD), red genes are super-up-regulated (SU), other genes are depicted in gray. (C) Venn diagrams indicating the total number of SC, SU and SD genes in ADC and SQCC cohorts. (d-e-f) (Top panels) Distributions of gene expression levels (normalized counts) for three representative genes (the SC gene CAPS in (d), the SU gene ESPRP1 in (e), the SD gene RILPL2 in (f)) in the ADC cohort (yellow), in the SQCC cohort (purple), and for the control patients (gray). The dashed lines represent the mean expressions. (Bottom panels) The corresponding percentages of patients deregulated for each shown gene in ADC and SQCC cohorts are represented by bar plots: gray for non-deregulated patients, blue for down-regulated patients and red for up-regulation patients.

proportion of genes are committed to specific deregulation patterns that occur in all patient of a given cohort. Given their specificities, the 'super' genes will likely be of interest in therapeutic research.

**Individual genetic deregulations efficiently classify cancer histology and identify novel adenocarcinoma molecular subtype.** ADC and SQCC histologies differ in gene expression. To assess the power of PenDA method compared to traditional analyses on normalized expression counts, we applied principal component analysis (PCA) on both PenDA differential expression matrix (values equal to -1 if a gene in a given tumor is down-regulated, 0 if a gene is not deregulated or 1 if a gene is up-regulated) and normalized count matrix (normalized RNA-seq counts with values between 0 and $3,7.10^6$ counts). In both cases, we observed a separate clustering of ADC and SQCC cohorts mainly driven by the first principal component (Fig 6A and 6B). We used a supervised learning algorithm (SVM, see Methods) to compare classification properties of *normalized count* versus *differential expression* inputs. Both approaches succeed to properly classify patients between ADC and SQCC histologies, though we observed that classification based on PenDA inputs performed slightly better (Fig 6C). We then applied hierarchical clustering to classify the 455 ADC and 473 SQCC samples together, using a subset of 875 genes defined in a previous independent study (based on RNA-seq counts) as lung cancer subtypes classifiers (Classification to Nearest Centroid, [40]). We clustered samples with a distance based on inter-sample Pearson correlations computed from the PenDA differential expression matrix (Fig 6D). We observed a clear separation between ADCs and SQCCs groups, thereby validating our methodological approach. We could identify one main SQCC class and three ADC subclasses (S3 Table). The majority of ADC patients clustered into 2 subclasses (class II and III), that were not distinguishable in the clustering analysis performed by George et al on different lung cancers, using the same classifier genes [40]. We compared the three ADC subclasses obtained with our approach with the six ADC genomic subtypes previously identified by Chen et al, using a multiplatform-based approach on the TCGA-LUAD dataset [27]. Class II ADC patients are mainly associated with AD1, AD2 and AD3 subtypes, whereas the majority of class III ADC patients is distributed among AD4 and AD5 subtypes (Fig 6E). Similarly, class II and class III ADC patients did not directly relate to the integrated ADC molecular subtypes defined by the pioneer work of The Cancer Genome Atlas Research Network [24] (S8A Fig). Interestingly, the same hierarchical clustering analysis using the same genes but with normalized counts did not clearly highlight the three ADC subtypes identified with the PenDA differential expression matrix (S8B Fig). Thus, clustering ADC according to their individual deregulation profiles identified new ADC subclasses. This demonstrates that personalized analysis using PenDA method brings new insights into histology classification.

**Systematic up-regulation of 37 genes in adenocarcinoma is a strong predictor of poor prognosis.** We then wondered what defined these novel ADC subclasses. First, we asked whether this segmentation into three classes was specific to the classifier genes chosen to perform the hierarchical clustering. We performed a principal component analysis on ADC

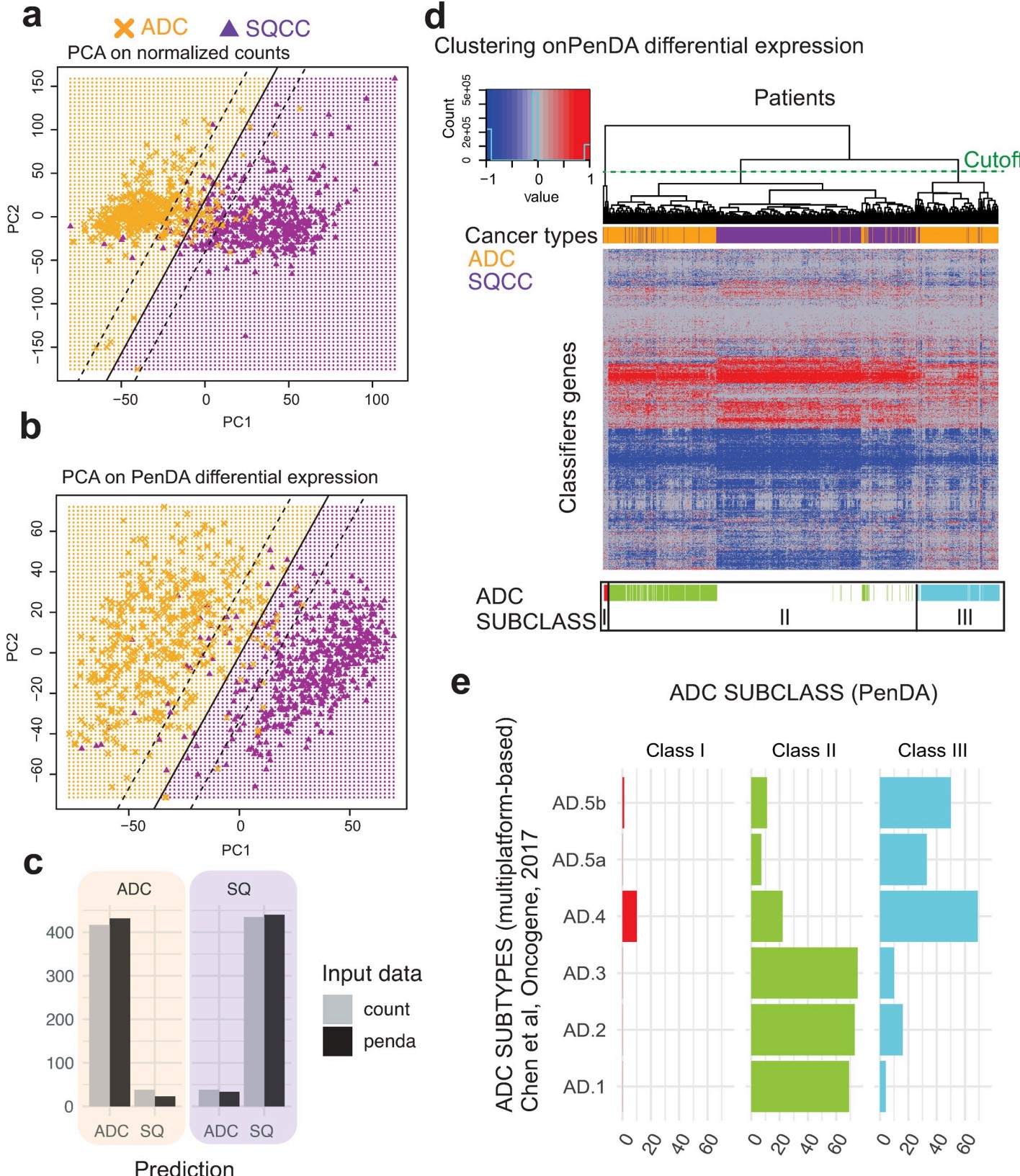

**Fig 6. Genetic deregulations efficiently classify cancer histologies.** (a, b) Principal Component Analysis on TCGA non-small-cell lung cancers (ADC and SQCC cohorts) using normalized count matrix (a) or PenDA differential expression matrix (b) as input. Full lines represent the decision boundary between ADC and SQCC histologies (using a linear SVM classifier on the first two principal components). Dashed lines represent the upper and lower margins of the decision boundary. Each symbol represents an individual sample (orange crosses for ADC, purple triangles for SQCC). (c) At the bottom, the bar plot represents the histology predictions based on the SVM classifier. SVM on PenDA predicts correctly 95% of ADCs and 93% SQCCs. SVM on count predicts correctly 92% of ADCs and 92% SQCCs. (d) Heatmap of PenDA differential expression matrix applied to a specific set of classifier genes (n = 875) in TCGA non-small-cell lung cancers: ADC (orange) and SQCC (purple). Two hierarchical clustering analyses were performed: using Euclidean distance to sort genes and using Pearson correlation-based distance to classify patients, with a complete linkage function in both cases. ADC subclasses (color-coded, class I to III) are defined according to the dendrogram cutoff n = 3 groups (cutting section = green dashed line). (e) Graphical representation of the contingency table between ADC subtypes (Chen et al,) and ADC subclasses (PenDA analysis). Each bar plot represents the total number of patients in each cell of the table.

cohort only using the corresponding PenDA differential expression matrix for all genes (Fig 7A). The first two principal components of the analysis nicely discriminated classes I, II and III. We then focused on the two major groups: class II and class III. We performed a Cox survival analysis on these two groups (Fig 7B) and observed that the class III patients have a better 5-year survival prognosis than class II patients (cox p-value = 0.00104). In order to better understand the molecular differences between class II and class III patients, we analyzed the pattern of deregulation of all genes in each class (Fig 7C). In class II, we observed a significant augmentation in the proportion of tumors where a given gene was detected as deregulated. In total, ~13% of the genes (n = 2432) were significantly more often deregulated in class II compared to class III patients (one-sided proportion test). We verified that the cancer stages, gender, and age were evenly distributed in class II and class III patients (chi square test p-value = 0.2133, p-value = 1, and p-value = 0.2133, respectively) and that the shift in genetic deregulation was detectable independently of stages, gender and age (S9 Fig). This indicated that this adenocarcinoma classification was not correlated with any of these putative confounding factors.

We decided to specifically study the 37 genes displaying the most extreme differences between the two classes, i.e. the genes deregulated in more than 75% of class II patients and in less than 25% of class III patients (red dots on Fig 7C, S4 Table). Since all these genes are committed toward up-regulation in class II patients, we tested if the up-regulation of these genes would be a good predictor of cancer survival. We added up the level of individual deregulation of the 37 genes (values equal to -1, 0 or 1, for each gene) to quantify the total deregulation score associated with those genes. Then we defined three groups using the $1^{st}$ and the $3^{rd}$ quantile of the score distribution. Analysis of the 5-years survival curve in the ADC LUAD-TCGA dataset showed a significant difference between groups, with a worst prognosis for patients that display up-regulation of most of the genes (score $\geq$ 34, Fig 7D). To validate our selected set of 37 genes as robust biomarkers, we applied the PenDA method on expression data (Affymetrix Human Genome U133 Plus 2.0 Array) of an independent adenocarcinoma cohort from the Grenoble Hospital (85 patients, GSE30219[4]) (see Methods). We then investigated the 5-years survival curve of the three groups predicted using 36 genes (all genes were analyzed in the Grenoble Hospital cohort, except FAM72D not measured by the array). Coherently with the results observed in TCGA-LUAD ADC cohort, patients up-regulated for many genes (score $\geq$ 15) have a worst prognosis (cox p-value = $5.2.10^{-4}$, Fig 7E). Thus, using the PenDA method, we identified 37 biomarkers predicting a bad outcome when they are all up-regulated. Altogether, these results suggest that PenDA method is a powerful approach to discover new biomarkers in cancer.

## Discussion

The PenDA method provides a new rank-based approach to analyze personalized gene deregulation. The method outcompetes existing approaches to identified genetic deregulation at the

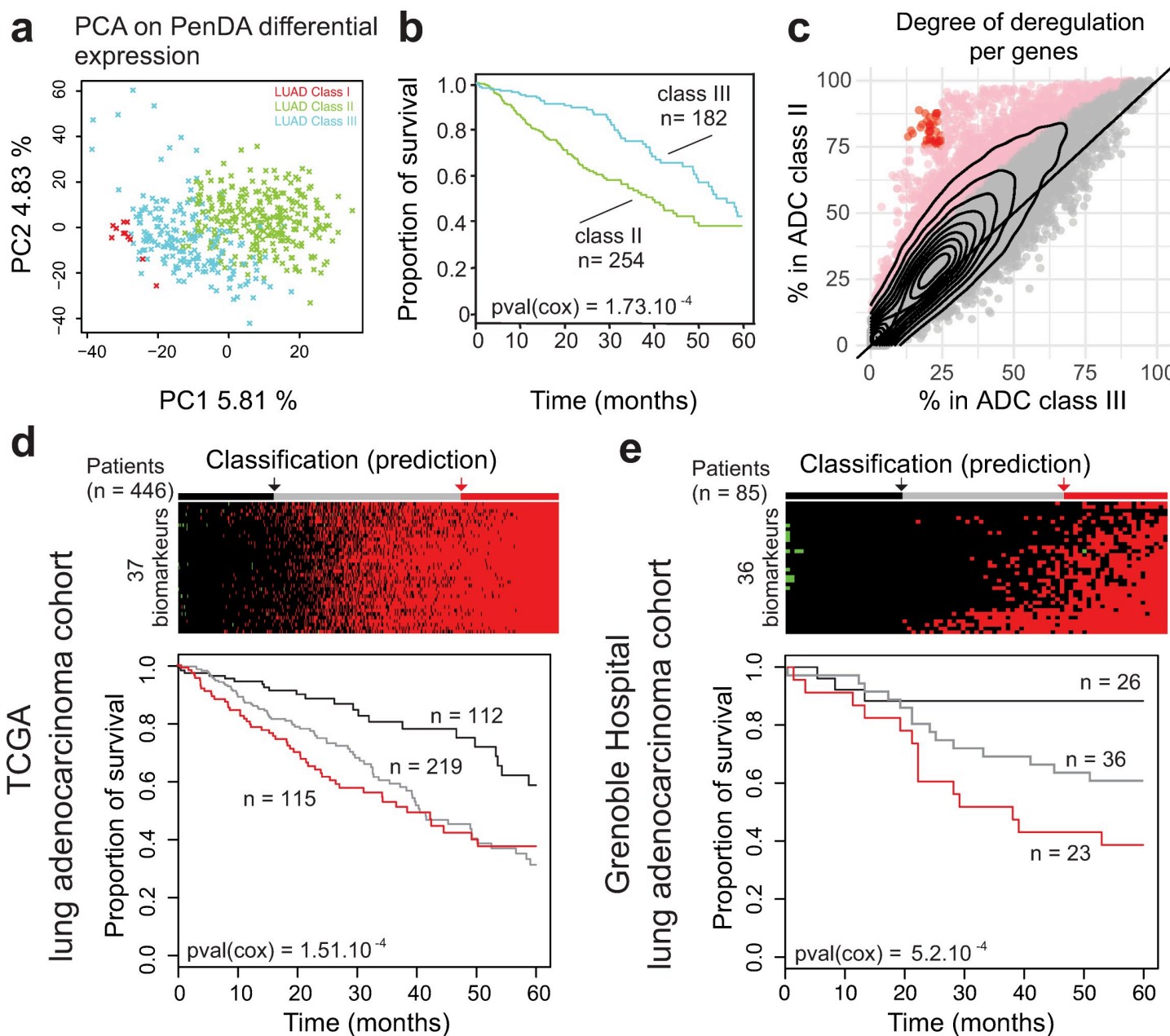

**Fig 7. Upregulation of 37 genes in adenocarcinoma is a strong predictor of poor prognosis.** (a) Principal Component Analysis on ADC cohort. Each cross represents an individual sample. The color of the dots represents the three subclasses defined in Fig 6. (b) Survival of ADC patients classified according to the 2 main subtypes (classes II and III). (c) The percentage of deregulated patients within the ADC class II (y-axis) or the ADC class III (x-axis). Each dot corresponds to one gene. The contour lines correspond to the density of genes. Pink dots indicate genes with a significant higher proportion of deregulation in the class II (proportion test, p-value < 0.05 after Bonferroni correction for multiple testing). Red dots define 37 genes highly deregulated (>75%) in the class II group and lowly deregulated (<25%) in the class III group. (d) (Top) Classification of ADC TCGA-LUAD built on the total number of up-regulated genes among the subset of 37 classifiers defined in (c). Patients are separated into 3 discrete groups: a group with a low upregulation (black, score < 4), a group with intermediate deregulation (gray, $4 \leq$ score < 34) and a group with most genes upregulated (red, $34 \leq$ score). (Bottom) Survival of patients according to these 3 groups. (e) As in (d) but for ADC Grenoble Hospital patients. Patients are separated into 3 discrete groups: a group with a low upregulation (black, score $\leq$ 0), a group with intermediate deregulation (gray, 0 < score < 15) and a group with most genes upregulated (red, $15 \leq$ score).

individual level on simulated datasets. Applied to non-small-cell lung cancer expression data, our method showed that gene deregulation varies in a continuous manner between patients. When frequently deregulated, genes tend to commit to specific deregulation patterns (up or

down regulation). We observed that a small proportion of genes exhibits unusual 'super' deregulation pattern (always down, up or non-deregulated). Personalized differential analysis succeeds to properly cluster adenocarcinoma and squamous cells lung cancer histology. More specifically, clustering analysis leads to the identification of 37 biomarkers that efficiently predict 5-years survival in two independent adenocarcinoma cohorts. The method is available as an open source R package called *penda*. We provide user guidelines so that *penda* could be installed and run by users with limited computational experience. To ensure reproducibility of analysis, the *penda* vignette provides a summary of used parameters ready to be included in the method section of publications using PenDA.

PenDA is robust against different techniques of transcriptome analysis and against batch effects. Notably, the biomarkers that we identified on the ADCs TCGA cohorts based on an RNA-seq technology was validated on an independent ADC cohort where gene expressions were measured with microarrays. Another advantage of the method is that it is easily generalizable to other types of data like transcript expression, DNA methylation, proteomics, etc. For instance, several methods have been recently developed and benchmarked for the inference of isoform abundance from RNA-seq data [46]. However, classical differential expression analytical tools (on RNA-seq count data) are based on gene features and are not optimized for the estimate of transcripts abundance data. Thus, testing for individual differential isoform abundances with PenDA would be an interesting challenge. The PenDA approach could also be adapted for single cell analysis [47] to leverage the understanding of single cell expression and to quantify intra-sample heterogeneity at the single cell level.

The current PenDA method has however several limitations. First, though our method does not depend on replicates to identify individual deregulation, it relies on a control cohort that is supposed to reliably define a 'normal' ranking. Therefore, it is crucial to properly define suitable control datasets. Second, PenDA individual expression analysis requires the use of genome-wide transcriptomic data. In the future, we would like to explore the possibility to define a set of super conserved genes that could serve as internal reference for 'partial' PenDA analysis on sparse qPCR data. Third, our method is not suitable for genes with low expression levels in all samples, which are currently removed by filtering in the first step of the analysis.

The aim of population differential analysis is to detect consistently up or down regulated genes, *in average*. The PenDA method was based on the concept that individual level analyses are complementary of population approaches. Applying DESeq2, one of the most common DE analysis software, to the ADC and SQCC TCGA cohorts, highlighted similarities and differences for the genes with specific deregulation patterns identified by PenDA (super-conserved, super-up-regulated, super-down-regulated) (S10A and S10B Fig). For example, if all SU and SD genes were identified as differentially regulated by DESeq2 at the population level, many genes detected by DESeq2 as deregulated with a large fold-change and a low adjusted p-value are deregulated only in a limited subset of patients. Moreover, PenDA provides a unique way of identifying genes that are significantly never de-regulated (super-conserved), a category of genes hardly detectable by population methods. Similarly, compared to another meta-analysis of genetic deregulation at the population level in non-small-cell lung cancer based on microarray gene expression data [48], we observed that none of the three super-up genes common between SQCCs and ADCs (*PAFAH1B3*, *CBLC* and *ESRP1*) were identified as up-regulated by Tian et al, and only 28 of the 128 super-down-regulated genes common between SQCC and ADC were identified as down-regulated in the same study (S10C Fig). More surprisingly, *CD19* and *IL10*, two genes involved in the immune response and never deregulated in SQCCs and ADCs TCGA cohorts were identified as over-expressed by Tian *et al*. These comparisons suggest that applying the PenDA approach and identifying individual genetic deregulation patterns can bring new, complementary insights into the comprehensive analysis

of non-small-cell lung cancers or other types of cancers. In particular, genes displaying a 'super' profile can be considered as generic candidates for therapeutic strategies.

The PenDA method generates useful individual information that can be incorporated into further functional analysis. With PenDA, we provided generalized statistics at the level of a single individual/sample and at the level of a single gene (number of deregulated genes per tumor, number of tumors where a gene is deregulated, proportion of up-regulation if differentially regulated, etc.). At the gene level, this individual information can be combined to increase the power to detect significant association with phenotypic outcome, such as survival. As an illustration, we analyzed the synergic effect of gene deregulation of the GINS complex on survival, in the ADC cohort. GINS is a four-genes complex essential for initiation and elongation during DNA replication [49]. High expression of this complex has been related to tumorigenic properties [50]. ADC patients are heterogeneously deregulated for each of the GINS complex member, we classified them into three groups, based on PenDA differential analysis: (-): absence of gene deregulation for all the 4 constitutive genes; (+): 1 to 3 gene deregulations; and (+++): all genes are simultaneously deregulated (Fig 8A). Overall survival of ADC patients could be significantly discriminated using the synergic effect of GINS deregulation (Fig 8B), however, no significant effect of GINS deregulation could be identified using single gene Cox regression models (Fig 8C). This example demonstrates the interest of exploring possible synergic effects of single gene deregulations, in each individual. Besides survival analysis, single gene differential analysis could be profitably included into network analyses [51] to identify driver genes and functional communities. Moreover, a systematic exploration of the relationship between driver mutations [52] and individualized expression deregulations is a promising strategy to improve the accuracy of future pan-genomic studies.

## Methods

### Data and preprocessing

Two datasets of gene expression (HTSeq-Counts) were downloaded from The Cancer Genome Atlas program (https://portal.gdc.cancer.gov/). The datasets contain tumor ('01' barcoded

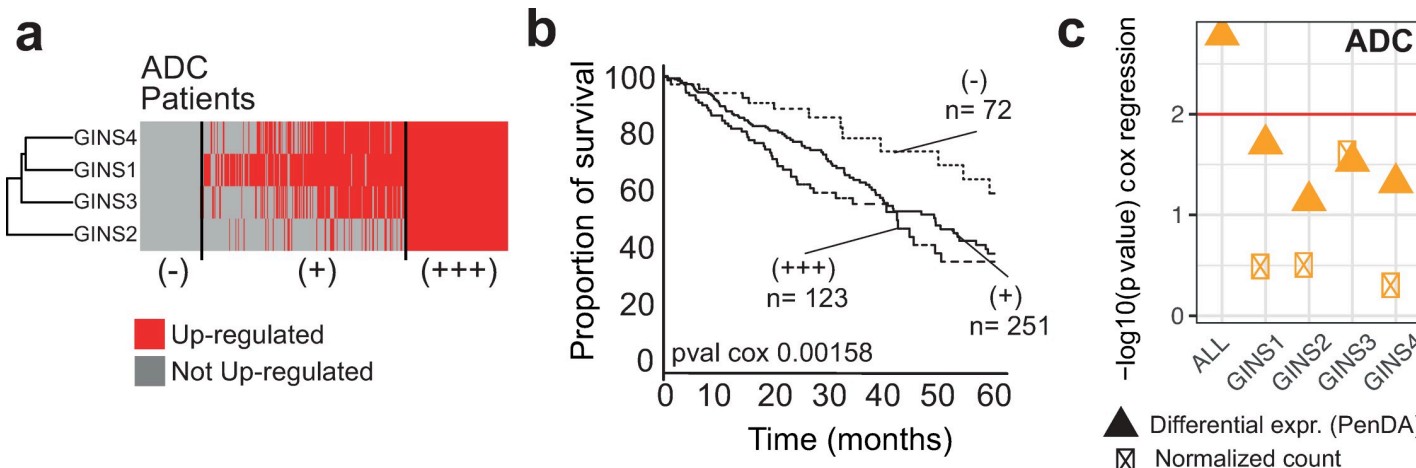

**Fig 8. Synergic effects of gene deregulation within a protein complex.** (a) Heatmap showing the distribution of gene deregulations of genes coding for the GINS complex in the ADC cohort. Patients are ordered from left to right according to an increasing number of gene deregulations within the GINS complex. The patients were separated into discrete deregulation groups of: 0 up-regulation (-), 1–3 up-regulations (+) and 4 up-regulations (+++). (b) Survival of ADC patients according to the deregulation groups defined in (a). (c) Cox regression p-values associated with different models (multivariate and univariate). Cox regression is applied on PenDA deregulation matrix (triangles) or expression matrix (ticked boxes, normalized count values). ALL corresponds to a multivariate cox model including the four genes of the GINS complex. The red line corresponds to the significance level of 0.01.

samples) and control ('11' barcoded samples) tissues from two non-small cells lung cancers: lung adenocarcinoma (LUAD or ADC) and lung squamous cell (LUSC or SQCC). Patients with prior malignancies and replicated samples were removed from the analysis. We kept 1026 samples: 455 ADC tumors, 473 SQCC tumors and 98 control tissues consisting of normal adjacent lung tissue samples (50 from the ADC cohort, 48 from the SQCC cohort). For further analysis, we selected 19177 protein coding genes (hg38 reference genome). This corresponds to protein-coding genes of the base RefSeqGene (https://www.ncbi.nlm.nih.gov/refseq/rsg/). We then normalized the HTSeq-Counts using the *estimateSizeFactors* and the *count* functions of the DESeq2 package [38]. Finally, data were filtered to remove genes with null expression (counts = 0) in all samples (controls and tumors). At the end, we kept 18143 protein-coding genes.

## The PenDA method

For each gene *g*, we first determined the lists *L(g)* and *H(g)* of other genes whose expressions are respectively lower or higher than that of *g* in at least 99% of the normal samples. These lists were next restricted to the subsets of *l* genes that have a median gene expression in normal samples closest to the corresponding median level of *g*, with *l* a user-defined parameter controlling the maximal size of *L* and *H* lists.

For a given tumor sample *T*, the personalized differential analysis was then performed iteratively:

a. For each gene *g*, we compared its expression *E(g,T)* in the tumor *T* to the corresponding expression of genes present in the *L* and *H* lists. It allowed to defined four non-overlapping sets of genes:

$$L_d = \{g' \in L(g) \ \setminus \ E(g', T) < E(g, T)\}$$

$$L_u = \{g' \in L(g) \ \setminus \ E(g', T) > E(g, T)\}$$

$$H_d = \{g' \in H(g) \ \setminus \ E(g', T) < E(g, T)\}$$

$$H_u = \{g' \in H(g) \ \setminus \ E(g', T) > E(g, T)\}$$

$L_u \neq \emptyset$ or $H_d \neq \emptyset$ indicated than the relative ordering of *g* has changed in *T* compared to the control cases.

b. We considered that a gene *g* is deregulated in *T* if and only if

$$\left(\frac{|L_u|}{|L|} \geq h\right) \vee \left(\frac{|H_d|}{|H|} \geq h\right) \tag{1}$$

with |*X*| the cardinality of ensemble *X* and *h* a user-defined parameter defining the minimal proportion of genes in *L* or *H* whose relative ordering with *g* has changed. If Eq (1) is satisfied then *g* is considered as down-regulated or up-regulated if $|L_d|+|H_d|<|L|$ or $|L_u|+|H_u|<|H|$ respectively. In the cases where the *L* or *H* lists are empty, we used the percentile method (see below) to take the decision on the status of *g* in *T*.

c. After having scanned all the genes, we aimed to minimize the potential bias that observed changes of ordering is actually due to the deregulation of genes in the *L* or *H* lists. Thus, we excluded in every *L* and *H* lists all the genes that had been diagnosed as deregulated in step

(b), and reiterated steps (a), (b) and (c) until convergence of the list of deregulated genes (S1 Fig, blue line), or until a user-specified number of iterations had been reached. It often happens that the final iterations oscillate between two lists (S1 Fig, red line). In this case, the union of both lists is considered as the predicted set of deregulated genes.

## The percentile method

The percentile method consists in finding if the expression value of a gene in a test-sample is an outlier of the distribution of expression for the same gene within an ensemble of reference samples. More precisely, for each gene $g$, we determined $p_l$ and $p_u$ respectively the $x$ and *(100-x)* percentiles of the distribution of expression $E(g,S)$ for $g$ within the ensemble of normal samples *{S}*, where $x$, given in %, is a user-tunable parameter. Then, a gene $g$ in tumor sample $T$ with an expression $E(g,T)$ was considered as differentially expressed in that sample if $E(g,T)<p_l/f$ (down-regulation) or $E(g,T)>p_u*f$ (up-regulation), with $f≥1$ a user-defined factor allowing to expand the window of normal expression. A ROC curve analysis obtained by varying $x$ and based on the simulated datasets (see below) suggested that using a factor $f~1.2$ leads to an optimized diagnosis with this method (S2 Fig).

## Simulated datasets

We generated realistic simulated datasets from the ensembles of normal and tumorous samples of the LUAD and LUSC TGCA studies. We first ranked all the gene expression values in normal samples and pooled them into consecutive packets. Each packet $k$ contained 100 values of similar range $\{E(g_{k,1},S_{k,1}), E(g_{k,2},S_{k,2}),\ldots, E(g_{k,100},S_{k,100})\}$ with $E(g_{k,i},S_{k,i})$ the expression of gene $g_{k,i}$ in normal sample $S_{k,i}$. Then for each group, we computed the ensemble of expression differences in normal samples defined as $\Delta_n(k) = \{E(g_{k,i},S')−E(g_{k,i},S_{k,i}), 1≤i≤100$ and $\forall\ S'≠S_{k,i}\}$. Similarly, we defined the ensemble of expression differences between tumorous and normal samples as $\Delta_c(k) = \{E(g_{k,i},T)−E(g_{k,i},S_{k,i}), 1≤i≤100$ and $\forall$ tumor $T\}$. From the 5% and 95% percentiles of $\Delta_n(k)$, noted $p_5(k)$ and $p_{95}(k)$ respectively, we isolated the subset $\Delta'_c(k)$ of values in $\Delta_c(k)$ that are smaller than $p_5(k)$ or greater than $p_{95}(k)$. We assumed that $\Delta'_c(k)$ represents typical abnormal expression differences observed in cancer for the packet $k$ and that the ratio $r(k)$ between the number of elements in $\Delta'_c(k)$ and in $\Delta_c(k)$ is representative of the probability for a gene in this group to be deregulated.

Finally, to generate a simulated tumorous sample, we chose randomly one normal sample $S$. For each gene $g$, we determined the packet $k$ containing $E(g,S)$ and its expression was modified with a probability $r(k)$ by adding a randomly-chosen element of $\Delta'_c(k)$. In average 30% of the genes were up or down-regulated. Instead of $r(k)$, we also used fixed proportions of deregulated genes from 0.05 to 0.9. We tested that the performance of PenDa on simulated datasets was not affected by the packet size (S11A Fig). The choice of the percentiles (5%, 95%) impacts on the ROC curves while PenDA still remains the best investigated methods in the low FPR range (S11B Fig).

Note that such strategies may be adapted to any data to generate realistic simulated datasets adapted to the user-defined system of interest.

## Predictive power on simulated datasets

To test the efficacy of PenDA or of other methods, we generated 10 simulated tumors (see above). For each dataset, in order to realize a fair comparison, we excluded the normal sample from which it was generated to the ensemble of normal samples used to define the reference

properties of each method. For a given method and given parameters, true positive (TPR), false positive (FPR) and false discovery (FDR) rates were computed on these 10 simulations. ROC curves (TPR vs FPR) were obtained by varying one specific parameter for each method (threshold $h$ for PenDA, percentile $x$ for the percentile method, FDR level for Rankcomp and log2 fold change threshold for DESeq2). From each curve, we extracted the maximal informedness defined as the maximal value of the Youden's $J$ statistics defined as the difference between TPR and FPR (TPR-FPR). An ideal predictive method would reach a maximal informedness of 1 while a random-decision method would approach 0 value.

In Fig 2, the effect of the number of control samples in the reference dataset (Fig 2B) and of the number of investigated genes (Fig 2C) were analyzed by randomly choosing a set of control samples or a set of genes from the initial pools and by repeating these operations 10 times. TPR and FPR levels were computed on the ensemble of simulations and of random choices. In Fig 3B, the ROC curves were determined for a set of 10 control samples randomly picked from the original pool. In Fig 3C, effect of normalization was simulated by multiplying RNA-seq counts of control and tumorous samples by random factors uniformly drawn between 1 and 5: the same factor was applied for all the genes of a given sample.

## Estimation of the false discovery rate of RankComp from results given in Wang et al

In their original paper [15], Wang et al performed simulations to test the RankComp method. Each simulated sample contains $T = 15000$ genes including $P = 3000$ deregulated genes. In Table 2 of [15], they gave the sensitivity $SE$ and specificity $SP$ of the method for several simulations. From that, we can compute the corresponding false discovery rate $FDR = (T-P)(1-SP)/[(T-P)(1-SP)+P*SE]$. Using this formula, the computed $FDR$s ranged from 20% to 50%.

## PenDA analysis of the lung cancer cohort from the TCGA

The PenDA method was applied on preprocessed expression TCGA data (see Methods section: 'Data and preprocessing'). The PenDA vignette of the penda package version 1.0 was executed on 18143 genes, using 98 control samples and 928 case samples. The data set was pretreated as following: 0 gene and 0 sample were removed during the NA values filtering step, and 1034 gene was removed for low because lowly expressed: under the threshold 'val_min' = 10 in at least 99% of cases. 98 controls were used to generate L and H lists using the following parameters: threshold LH = 0.99 and s_max = 30. The penda method was then applied on 928 cases, with the following set of parameters: quantile = 0.02, factor = 1.2 and threshold = 0.3.

## PenDA analysis of the lung cancer cohort from the Grenoble Hospital

The PenDA method was applied on expression data (Affymetrix Human Genome U133 Plus 2.0 Array) of the GSE30219 cohort. The PenDA vignette of the penda package version 1.0 was executed on 19148 genes, using 14 control samples and 293 case samples. The data set was pretreated as following: 0 gene and 0 sample were removed during the NA values filtering step, and 0 gene was removed for low because lowly expressed: under the threshold 'val_min' = 0.5 in at least 99% of cases. 14 controls were used to generate L and H lists using the following parameters: threshold LH = 0.99 and s_max = 100. The penda method was then applied on 293 cases, with the following set of parameters: quantile = 0.05, factor = 1.05 and threshold = 0.8.

## Statistical analyses

Statistical analyses were performed on the following PenDA deregulation matrices, for $S$ samples (tumors) and $G$ genes:

- The upregulated matrix $U_{mat}$ with $U_{mat}(g,T) = 1$ if gene $g$ is up-regulated in tumor $T$ (= 0 otherwise), with $T \in (1, \ldots, S)$ and $g \in (1, \ldots, G)$

- The downregulated matrix $D_{mat}$ with $D_{mat}(g,T) = 1$ if gene $g$ is down-regulated in tumor $T$ (= 0 otherwise), with $T \in (1, \ldots, S)$ and $g \in (1, \ldots, G)$

- The matrix of total deregulation $Tot_{mat} \equiv U_{mat} + D_{mat}$.

a. Testing for equality of deregulation proportions (Fig 4) was performed using two-sided two-proportion z-test (prop.test function in R), with a Bonferroni corrected p-value threshold at $2.75.10^{-6}$ (corresponding to 18143 multiple testing).

b. Statistically significant deregulation frequency (Fig 5) was assessed by a t-statistic computed for each gene. The t-statistic was calculated using the R t.test function, with the vector of $S$ values corresponding to the estimated differential expression $x_{gT}$ for the gene $g$ in each tumor $T$ and the true value of the mean defined as $\left\{ mu = \frac{1}{G} \sum_{g=1}^{G} (\frac{1}{S} \sum_{T=1}^{S} x_{gT}), \ x \in \{0, 1\} \right\}$.
A calibrated p-value associated with the t-statistic and a corresponding q-value were then calculated using the R package fdrtool using the following parameters: cutoff.method = "pct0" and pct0 = 0.90 [53].
The test was applied on the $Tot_{mat}$. Super-up-regulated genes were defined as follows: (i) $\sum_{T=1}^{S} x_{gT}^{U} > median(\sum_{T=1}^{S} x_{T}^{Tot})$, ii) counts > 10 in at least 80% of the control samples and iii) significant t.test q-value. Super-down-regulated genes were defined as follows: (i) $\sum_{T=1}^{S} x_{gT}^{D} > median(\sum_{T=1}^{S} x_{T}^{Tot})$, ii) counts > 10 in at least 80% of the control samples and iii) significant t.test q-value. Super-conservde genes were defined as follows: (i) $\sum_{T=1}^{S} x_{gT}^{Tot} < median(\sum_{T=1}^{S} x_{T}^{Tot})$, ii) counts > 10 in at least 80% of the control samples and iii) significant t.test q-value.

c. PCA analysis (Fig 6) was performed using the function big_randomSVD of the R package bigstatr[54]. SVM linear regression was performed on the 2 firsts Principle Component of PCA analysis, using the function svm of the R package "e1071", using the following arguments: kernel = linear, cost = 10 and scale = FALSE.

## Survival analyses

The R package survival was used to compute Cox-models and create 5-years survival curve (Fig 6 and Fig 7). The *survival::coxph* function was used to fit a Cox proportional hazard regression model and the overall likelihood ratio p-value was extracted for further analysis. The *survival::survfit* function was used to create survival curves from the Kaplan-Meier estimate.

## Gene functional classification

Gene functional classification was performed using the DAVID's Functional Annotation tool of David Bioinformatics Resources 6.8 [55,56]. Enrichment analyses for gene lists of interest were performed against Gene Ontology term–Biological Pathway (direct) repository.

Heatmaps summarizing the results were generated from Functional Annotation Chart, after applying a cutoff of 0.001 on the Modified Fisher Exact P-Value (we used the tutorial kindly provided by Kevin Blighe).

## Use of Rankcomp

The original Rankcomp and the RankcompV2 algorithms [15,16] were tested using the Relative Expression Ordering Analysis (REOA) package downloaded from https://github.com/pathint/reoa. We ran the program *reoa* on our simulated datasets using the options *–s 1 –j 2 –a 2* to get individual predictions for both algorithms with default parameters. Results for different FDR levels were obtained using the *–f* option.

## Use of DESeq2

The R-package of DESeq2 [7] was imported from Bioconductor3.7. To assess fold changes in expression from simulated datasets, we used DESeq2 default parameters. We performed 10 comparisons between individual simulated tumor sample and 97 independent TCGA control samples (we remove the control sample used for simulating the tumor sample from the reference dataset). As no replicate was available for tumor sample, DESeq2 allowed the variance-mean dependence estimated from control samples to be used for case sample [57]. The log2-foldChange estimation was used for sensitivity and specificity analysis. Performing DESeq2 with or without its internal normalization routine may impact the ROC analysis in particular if data are not standardized (S11C Fig). To assess fold changes in expression from TCGA datasets, we applied DESeq2 methods with default parameters, except for the significance cutoff which was set to 0.01 (alpha value of the *DESeq2::results* function).

## Supporting information

**S1 Fig. Convergence towards a consistent list of deregulated genes is rapidly achieved by the PenDA method.** We plotted the evolution of the total number of predicted deregulated genes during the successive iterations of the PenDA method applied to one simulated dataset with $l = 30$ and $h = 0.1$ (red line) or $h = 0.4$ (cyan line).
(PDF)

**S2 Fig. Test of the percentile method.** (a) ROC curve of the percentile method obtained by varying parameter $x$ for different values of factor $f$. TPR and FPR were computed on a set of 10 simulations. (b) Maximal informedness of the ROC curve as a function of $f$.
(PDF)

**S3 Fig. PenDA workflow.**
(PDF)

**S4 Fig. Effect of tumor stages on gene deregulations for ADC and SQCC patients.** (a,b) Effect of tumor stages on the total number of gene deregulations in both ADC (a) and SQCC patients (b). (c,d) Effect of tumor stages on down/up ratios in both ADC (c) and SQCC patients (d). Significance was assessed via one-way ANOVA with Tukey's multiple comparison post hoc test, considering the stage as an independent factor, with 5 different levels (stage ia, stage ib, stage ii, stage iii and stage iv). LUAD gene deregulation: Df = 4, F-statistic = 5.18, p-value = 0.0004. LUAD deregulation ratio: Df = 4, F-statistic = 0.99, p-value = 0.4128. LUSC gene deregulation: Df = 4, F-statistic = 3.00, p-value = 0.0182. LUAD deregulation ratio: Df = 4, F-statistic = 1.59, p-value = 0.1767. Dashed red lines represent 1st quartile, median and

3rd quartile of the distributions.
(PDF)

**S5 Fig. Gene Ontology (biological pathways) enrichment.** GO (biological pathways) enrichment in genes significantly down in ADC compared to SQCC (a), significantly up in ADC compared to SQCC (b), significantly down in SQCC compared to ADC (c) and significantly up in SQCC compared to ADC (d). 1000 top hits of prop.test analysis were used to estimate terms enrichment in each condition. Rows of the heatmap correspond to genes overlapping with at least one enriched term (red). Genes with no overlapping terms were removed from the graphical representation. Columns correspond to enriched terms clustered by Euclidean distance. GO Term significance score corresponds to -log10 of the Modified Fisher Exact P-Value after Benjamini correction (extracted from DAVID's Functional Annotation tool).
(PDF)

**S6 Fig. Gene deregulation commitment in ADCs and SQCCs.** Genes deregulated in more than 30% of the patient are depicted in the diagram. x-axis corresponds to the % of up-regulation/total-deregulation in ADC, y-axis corresponds to the % of up-regulation/ total-deregulation in SQCC. Each dot (gray cross) corresponds to one gene. Blue points correspond to super-down-regulated genes, red points correspond to super-up-regulated genes (triangles for ADC, circles for SQCC). Diamonds black points represent genes displaying antagonistic commitment behaviour between ADC and SQCC (~5% of the total number of genes depicted).
(PDF)

**S7 Fig. Gene Ontology (biological pathways) enrichment in genes super-up-regulated and genes super-down-regulated.** GO (biological pathways) analysis for super-up-regulated (a) and super-down-regulated (b) genes in ADC or SQCC. Rows of the heatmap correspond to genes overlapping with at least one enriched term (red). Genes with no overlapping terms were removed from the graphical representation. Columns correspond to enriched terms clustered by Euclidean distance. GO Term significance score corresponds to -log10 of the Modified Fisher Exact P-Value after Benjamini correction (extracted from DAVID's Functional Annotation tool).
(PDF)

**S8 Fig. Comparison of ADC subclasses obtained from PenDA analysis with clustering analysis on normalized counts analysis and with ADC iClusters.** (a) Graphical representation of the contingency table between ADC iCluster (The Cancer Genome Atlas Research Network) and ADC subclasses (PenDA analysis). (b) Heatmap of normalized counts matrix applied to a specific set of classifier genes (n = 875) in TCGA non-small-cell lung cancers: ADC (orange) and SQCC (purple). Two hierarchical clusterings were performed: using Euclidean distance to sort genes and using Pearson correlation-based distance to classify patients, with a complete linkage function in both cases. ADC subclasses defined by PenDA analysis (colour-coded, class I to III) are defined according to Fig 6 of the main text.
(PDF)

**S9 Fig. Effect of putative confounding factors on ADC classification in class II and III.** (a) Effect of cancer stage patients (chi-square test p-value = 0.2133). (b) Effect of gender (chi square test p-value = 1). (c) Effect of age patients (chi square test p-value = 0.2133).
(PDF)

**S10 Fig. DESeq2 analysis of the ADC and SQCC TCGA cohorts.** DESeq2 analysis of the ADC (a) and SQCC (b) TCGA cohorts (green triangles: super-conserved genes, red triangles: super-up-regulated genes, blue triangles: super-down-regulated genes). (c) Genes identified as

deregulated by Tian et al. x-axis corresponds to normalized mean expression in controls. y-axis corresponds to normalized mean expression in tumor. Gene with super patterns identified with PenDA are depicted with triangles (green triangles: super-conserved genes, red triangles: super-up-regulated genes, blue triangles: super-down-regulated genes).
(PDF)

**S11 Fig. Effect of the simulation method parameters on PenDA performance.** (a) ROC curves (true positive rate TPR vs false positive rate FPR) of the PenDA method on different simulated datasets obtained with different packet sizes. The curves were obtained by varying the proportion threshold h. (b) Comparison with other methods. ROC curves on simulated datasets generated using percentiles 10%,90% (Left) and 20%,80% (Right) in the simulation method (see Methods in the main text) for PenDA, a simple percentile-based method, 2 versions of Rankcomp and DESeq2. (c) ROC curves of the full DESeq2 method (full lines) or with the DESeq2 method skipping the internal routine for normalization (dashed lines) for the same simulated dataset used in Fig 2 and Fig 3A of the main text (green) or with the non-standardized dataset used in Fig 3C of the main text.
(PDF)

**S1 Text. PenDA method vignette: Vignette_penda.**
(PDF)

**S2 Text. PenDA simulation vignette: Vignette_simulation.**
(PDF)

**S1 Table. Gene deregulation per sample.**
(CSV)

**S2 Table. Genetic deregulation profiles.**
(CSV)

**S3 Table. ADC clusters.**
(CSV)

**S4 Table. The list of 37 biomarkers.**
(CSV)

## Acknowledgments

We thank the members of the DJ's and SK's groups, in particular Ekaterina Flin, for inspiring discussions during regular joint group meetings. We are grateful to Florian Privé, Michael Blum, Eric Fanchon and members of the BCM team for algorithmic and methodological advices. We acknowledge computational resources from CIMENT infrastructure (supported by the Rhone-Alpes region, Grant CPER07 13 CIRA). EB thanks Centre de ressources (CRB) CHUGA Grenoble, French Ligue contre le cancer for transcriptomic platform and Nicolas Lemaitre for tumor data management.

## Author Contributions

**Conceptualization:** Magali Richard, Daniel Jost.

**Data curation:** Magali Richard, Clémentine Decamps, Florent Chuffart.

**Formal analysis:** Magali Richard, Clémentine Decamps, Daniel Jost.

**Funding acquisition:** Magali Richard, Elisabeth Brambilla, Saadi Khochbin, Daniel Jost.

**Investigation:** Magali Richard, Clémentine Decamps, Florent Chuffart, Elisabeth Brambilla, Sophie Rousseaux, Saadi Khochbin, Daniel Jost.

**Methodology:** Magali Richard, Clémentine Decamps, Daniel Jost.

**Project administration:** Magali Richard, Daniel Jost.

**Resources:** Magali Richard, Clémentine Decamps, Florent Chuffart.

**Software:** Magali Richard, Clémentine Decamps, Florent Chuffart.

**Supervision:** Magali Richard, Daniel Jost.

**Validation:** Magali Richard, Clémentine Decamps, Florent Chuffart, Elisabeth Brambilla, Sophie Rousseaux, Daniel Jost.

**Visualization:** Magali Richard, Clémentine Decamps, Florent Chuffart, Daniel Jost.

**Writing – original draft:** Magali Richard, Clémentine Decamps, Daniel Jost.

**Writing – review & editing:** Magali Richard, Daniel Jost.

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
