## [Decision Letter · Decision Letter 0]

25 Mar 2020

Dear Dr Richard,

Thank you very much for submitting your manuscript "PenDA, a rank-based method for Personalized Differential Analysis: application to lung cancer" for consideration at PLOS Computational Biology. As with all papers reviewed by the journal, your manuscript was reviewed by members of the editorial board and by several independent reviewers. The reviewers appreciated the attention to an important topic. Based on the reviews, we are likely to accept this manuscript for publication, providing that you modify the manuscript according to the review recommendations.

Sincerely,

Amin Emad

Guest Editor

PLOS Computational Biology

Jian Ma

Deputy Editor

PLOS Computational Biology

[LINK]

Reviewer's Responses to Questions

**Comments to the Authors:**

Reviewer #1: The authors presented a unique approach for identifying dysregulated genes in cancer data. The overall quality of the study, including writing, organization, and presentation is good. I wasn’t familiar with this type of approach, this manuscript does make a good introduction and I found it has the potential to be useful in future research studies.

Some minor comments

1. The authors summarized DEseq2 edgeR, limma to be fold-change method. These methods do provide fold change, but their algorithm is much more than just fold change.

2. Detail of how data was simulated should be provided.

3. Small grammar errors such as “Help to found”

Reviewer #2: In this interesting manuscript, Richard et al. introduce a new method, PenDA, for identifying genes that are deregulated in individual samples, based on comparison to a reference group of samples. The method works by comparing the ranking of each query gene relative to a set of genes that are consistently above or below the query gene in the reference group, and whose overall expression level is as close to the query gene as possible. The method is very simple, but in my opinion the authors have adequately shown that despite its simplicity, it performs quite well. The performance evaluation is almost entirely based on simulated data, but the simulations are performed carefully to take into account differences in behaviour of genes with varying baseline expression levels, and also to study the effect of various parameters that can be potentially tuned. The authors have then applied their method to TCGA lung cancer data, showing that their approach can identify differences between tumour and normal, can distinguish lung adenocarcinoma from squamous cell lung carcinoma, and can further identify potentially new subtypes within the adenocarcinoma cohort that correlate with genomic subtypes. Overall, I enjoyed reading the manuscript, and have only minor comments with respect to some of the details of the method evaluation and comparison:

1. In their simulations, the authors randomly choose a fraction of genes as deregulated, and modify the expression of these genes in a way that the amount of change in gene expression exceeds the natural variation that is seen in control (normal) samples (< 5th percentile or > 95th percentile). Does the choice of the percentile threshold affect the results? Can the authors perform simulations with at least one other percentile threshold (e.g. < 20th and > 80th percentiles) to show that the method performance is robust with respect to how the simulated data are constructed? Or if the performance is decreased, does the same apply to other methods that PenDA is compared to?

2. Among several different methods that the authors have evaluated, they test the ability of DESeq2 to identify deregulated genes in simulated data. My understanding from the Methods section is that the simulations are performed based on normalized data, and therefore the simulated dataset will also on the same normalized scale. However, DESeq2 requires raw read counts. Could the authors please clarify?

3. The authors show that the lung adenocarcinoma samples can be divided into three subtypes (classes I, II, and III) based on PenDA differential expression status. They state in the manuscript that a previous work could not identify these classes based on normalized counts, but it is not clear to me if this really shows the effect of PenDA vs. normalized counts, or if it reflects other potential differences in the analysis approach. Can the authors please repeat the same hierarchical clustering analysis using the same genes but with normalized counts, and compare to PenDA-based clustering?

**Have all data underlying the figures and results presented in the manuscript been provided?**

Reviewer #1: Yes

Reviewer #2: Yes

PLOS authors have the option to publish the peer review history of their article (what does this mean?). If published, this will include your full peer review and any attached files.

Reviewer #1: No

Reviewer #2: No
---

## [Decision Letter · Decision Letter 1]

11 Apr 2020

Dear Dr Richard,

We are pleased to inform you that your manuscript 'PenDA, a rank-based method for Personalized Differential Analysis: application to lung cancer' has been provisionally accepted for publication in PLOS Computational Biology.

Best regards,

Amin Emad

Guest Editor

PLOS Computational Biology

Jian Ma

Deputy Editor

PLOS Computational Biology

Reviewer's Responses to Questions

**Comments to the Authors:**

Reviewer #1: The authors have answered all of my comments.

Reviewer #2: The authors have adequately responded to my comments.

**Have all data underlying the figures and results presented in the manuscript been provided?**

Reviewer #1: Yes

Reviewer #2: Yes

PLOS authors have the option to publish the peer review history of their article (what does this mean?). If published, this will include your full peer review and any attached files.

Reviewer #1: No

Reviewer #2: No

---

## [Editor Report · Acceptance letter]

30 Apr 2020

PCOMPBIOL-D-20-00104R1 

PenDA, a rank-based method for Personalized Differential Analysis: application to lung cancer

Dear Dr Richard,

I am pleased to inform you that your manuscript has been formally accepted for publication in PLOS Computational Biology. Your manuscript is now with our production department and you will be notified of the publication date in due course.

With kind regards,

Matt Lyles
